# Tamoxifen blocks retrograde trafficking of Shiga toxin 1 and 2 and protects against lethal toxicosis

Andrey S Selyunin[1], Steven Hutchens[1], Stanton F McHardy[2], Somshuvra Mukhopadhyay[1]

**Shiga toxin 1 (STx1) and 2 (STx2), produced by Shiga toxin–producing *Escherichia coli*, cause lethal untreatable disease. The toxins invade cells via retrograde trafficking. Direct early endosome-to-Golgi transport allows the toxins to evade degradative late endosomes. Blocking toxin trafficking, particularly at the early endosome-to-Golgi step, is appealing, but transport mechanisms of the more disease-relevant STx2 are unclear. Using data from a genome-wide siRNA screen, we discovered that disruption of the fusion of late endosomes, but not autophagosomes, with lysosomes blocked the early endosome-to-Golgi transport of STx2. A subsequent screen of clinically approved lysosome-targeting drugs identified tamoxifen (TAM) to be a potent inhibitor of the trafficking and toxicity of STx1 and STx2 in cells. The protective effect was independent of estrogen receptors but dependent on the weak base property of TAM, which allowed TAM to increase endolysosomal pH and alter endosomal dynamics. Importantly, TAM treatment enhanced survival of mice injected with a lethal dose of STx1 or STx2. Thus, it may be possible to repurpose TAM for treating Shiga toxin–producing *E. coli* infections.**

## Introduction

Shiga toxin–producing *Escherichia coli* (STEC) infect more than 100,000 individuals each year and are a major cause of lethal food-borne infections (1, 2, 3). STEC produce two related toxins, Shiga toxin 1 (STx1) and 2 (STx2), which kill cells by blocking ribosomal protein synthesis (4, 5). Patients infected with STEC initially develop gastrointestinal disease (2, 3). In a subset (~5–15%), the toxins enter the bloodstream and cause life-threatening or fatal renal disease (2, 3). Definitive therapies are not available for STEC infections—there are no antidotes for STx1 and STx2, and antibiotic therapy is contraindicated because it may increase toxin release from STEC (2).

STx1 and STx2 are formed by the association of an A subunit, which is catalytically active, with a pentameric B-subunit, which mediates retrograde intracellular trafficking (4, 5, 6, 7, 8, 9). Retrograde transport of both toxins involves, sequentially, endocytosis, transit through early endosomes and the Golgi apparatus, and delivery to the endoplasmic reticulum from where the A subunit is translocated to the cytosol (5, 6, 7, 8, 9). Direct transport from early endosomes to the Golgi is critical as it allows the toxins to evade late endosomes where proteolytic enzymes are active (5, 6, 7, 8, 9). As STx1 and STx2 must traffic to the cytosol to induce cytotoxicity, blocking toxin transport in general, and at the early endosome-to-Golgi step in particular, has emerged as a promising therapeutic strategy (5, 6, 10, 11). As an example, treatment with manganese degrades the endosomal STx1 receptor GPP130 and thereby blocks the early endosome-to-Golgi transport of STx1, diverts STx1 to late endosomes for degradation, and protects cells and mice against lethal STx1 toxicosis (6). However, in order to be therapeutically effective, a toxin transport inhibitor must block STx2 because STx2 is ~400 times more toxic than STx1 in vivo (12), and in humans, disease severity correlates with STx2 production (13). In spite of the greater disease relevance, molecular mechanisms of STx2 transport, which is GPP130 independent and manganese insensitive (7), are poorly understood. This gap in knowledge has hindered therapeutic development, and currently, there are no toxin transport inhibitors approved for use in humans.

Here, we utilize data from a genome-wide siRNA screen and report the unexpected finding that early endosome-to-Golgi transport of STx2 requires efficient fusion of late endosomes with lysosomes. Inhibition of late endosome–lysosome fusion alters endosomal recruitment of retromer, which is required for the early endosome-to-Golgi transport of STx2 (9), providing a possible explanation for the effects on toxin trafficking. Through a subsequent screen of clinically approved drugs that target lysosomes, we identify tamoxifen (TAM) to be a potent inhibitor of the early endosome-to-Golgi transport and toxicity of STx2 and STx1. Further, we show that TAM acts as a weak base to increase endolysosomal pH, which alters endosomal dynamics and impacts endosomal recruitment of retromer. Finally, we show that TAM increases the survival of mice exposed to lethal STx2 or STx1. These findings identify a previously unknown role of late endosome–lysosome fusion in cargo transport at the early endosome/Golgi interface. Moreover, our work suggests that it may be possible to repurpose TAM for treating STEC infections.

[1]Division of Pharmacology and Toxicology, College of Pharmacy; Institute for Cellular and Molecular Biology; and Institute for Neuroscience, The University of Texas at Austin, Austin, TX, USA   [2]Center for Innovative Drug Discovery, Department of Chemistry, University of Texas San Antonio, San Antonio, TX, USA

Correspondence: som@austin.utexas.edu

# Results

### Biogenesis or function of lysosomes and/or autophagy is required for STx2 transport and toxicity

To elucidate the mechanisms of STx2 trafficking, we recently performed a viability-based genome-wide siRNA screen and identified 12 endosome/Golgi-localized host proteins that, when depleted, reproducibly protected against STx2-induced cell death (8). Surprisingly, 6 of 12 identified hits (Rab2a, FUT1, STAM, TPCN1, SNX14, and VEGFR2) regulate lysosome biogenesis/function and/or autophagy (Table 1). Based on this, here we hypothesized that biogenesis or function of lysosomes and/or the autophagy pathway is required for the trafficking and toxicity of STx2, and targeting lysosomes/autophagy may provide a therapeutically viable means to block STx2 trafficking.

To test this hypothesis, we first focused on one hit, Rab2a, and generated a stable HeLa cell clone in which Rab2a was depleted using a lentivirus-based CRISPR/Cas9 system. In the generated ΔRab2a clone, two separate stop codons were introduced in Rab2a, indicative of independent mutations in two chromosomes, and Rab2a transcript was not detectable (Fig 1A and B). Lysosomes fuse with late endosomes or autophagosomes to degrade endocytic or autophagic cargo, respectively (29, 30). The cytosolic protein LC3 is recruited to autophagosomes and degraded after autophagosome–lysosome fusion (31). ΔRab2a cells had a higher number of LC3-positive punctae than WT cells (Fig 1C and D), indicating that autophagy and/or lysosome function was compromised. Toxin transport assays revealed that, consistent with our previous studies (8, 9), in WT cells, STx2 B-subunit (STx2B) bound the cell surface and trafficked to the Golgi within 60 min (Fig 1E and F). In ΔRab2a cells, STx2B also bound the cell surface, but at 60 min, a pool of the toxin failed to traffic to the Golgi and instead was degraded (Fig 1E and F). At earlier time points, in ΔRab2a cells, STx2B was detected in Rab5-positive punctae (Fig 1G), indicating that internalization to early endosomes was not affected. Degradation of STx2B in ΔRab2a cells

was blocked by pretreatment with leupeptin/pepstatin or expression of dominant negative Rab7 (Fig 1H–K), suggesting that the toxin was degraded in late endosomes/lysosomes. Toxin degradation in ΔRab2a cells, in spite of possible changes in lysosomal function, was not surprising because soluble cargo are effectively degraded in prelysosomal late endosomes, where proteolytic enzymes are active (32). The block in transport was rescued by overexpression of WT, but not dominant negative or constitutively active, Rab2a (Fig 1L and M). Identical results were obtained using a second clone in which the CRISPR/Cas9 system introduced a stop codon in one Rab2a allele and an inactivating point mutation in the other (Fig S1A–F). Moreover, dicer-mediated knockdown of two other hits, STAM or FUT1, enhanced LC3 punctae, blocked endosome-to-Golgi transport of STx2B, and induced STx2B degradation (Fig S2A–E). Finally, we had previously demonstrated that UNC50, another hit on our screen, mediated early endosome-to-Golgi transport of STx2B by recruiting the ARF-GEF GBF1 to Golgi membranes (8). Analyses of cells lacking UNC50 or depleted in GBF1 revealed enhanced LC3-positive punctae as well (Fig S2F and G). Thus, depletion of four separate proteins (Rab2a, STAM, FUT1, or UNC50) blocked trafficking of STx2B to the Golgi and also impacted lysosomes and/or autophagy, bolstering the hypothesis that formation/function of lysosomes and/or autophagy is itself required for toxin transport.

### Fusion of late endosomes with lysosomes is necessary for the transport of STx2 from early endosomes to the Golgi, but the autophagy pathway is dispensable

To directly test the above hypothesis and distinguish between the role of lysosomes and autophagy, we took advantage of the fact that the HOPS tethering complex is required for the fusion of both late endosomes and autophagosomes with lysosomes (33, 34, 35). Depletion of Vps39, a component of the HOPS complex, blocks both these membrane fusion events and inhibits lysosome biogenesis/function as well as autophagy (33, 34, 35). In contrast, formation of

**Table 1. Role of TPCN1, Rab2a, SNX14, STAM, VEGFR, and FUT1 in lysosome function and/or autophagy.**

| Hit | Role in lysosome function/autophagy | Reference |
|---|---|---|
| TPCN1 | Endosome-localized calcium channel required for autophagy and lysosome maturation | (14) |
| Rab2a | Small GTPase historically associated with transport between the endoplasmic reticulum and the Golgi apparatus. Recent studies show that Rab2a also localizes to the endolysosomal system and is required for fusion of both late endosomes and autophagosomes with lysosomes. | (15, 16, 17, 18, 19, 20, 21) |
| SNX14 | Sorting nexin. Depletion leads to formation of enlarged lysosomes and accumulation of autophagosomes. | (22) |
| STAM | Part of the ESCRT-0 complex, which is required for the degradation of ubiquitylated proteins in lysosomes and formation of multivesicular endosomes. ESCRT-0 also plays a role in autophagy. | (23, 24, 25, 26) |
| VEGFR2 | VEGFR2 signaling induces autophagy | (27) |
| FUT1 | Mediates fucosylation of the lysosomal membrane proteins Lamp1 and Lamp2. Depletion inhibits fucosylation of Lamp proteins and alters autophagy and subcellular distribution of lysosomes. | (28) |

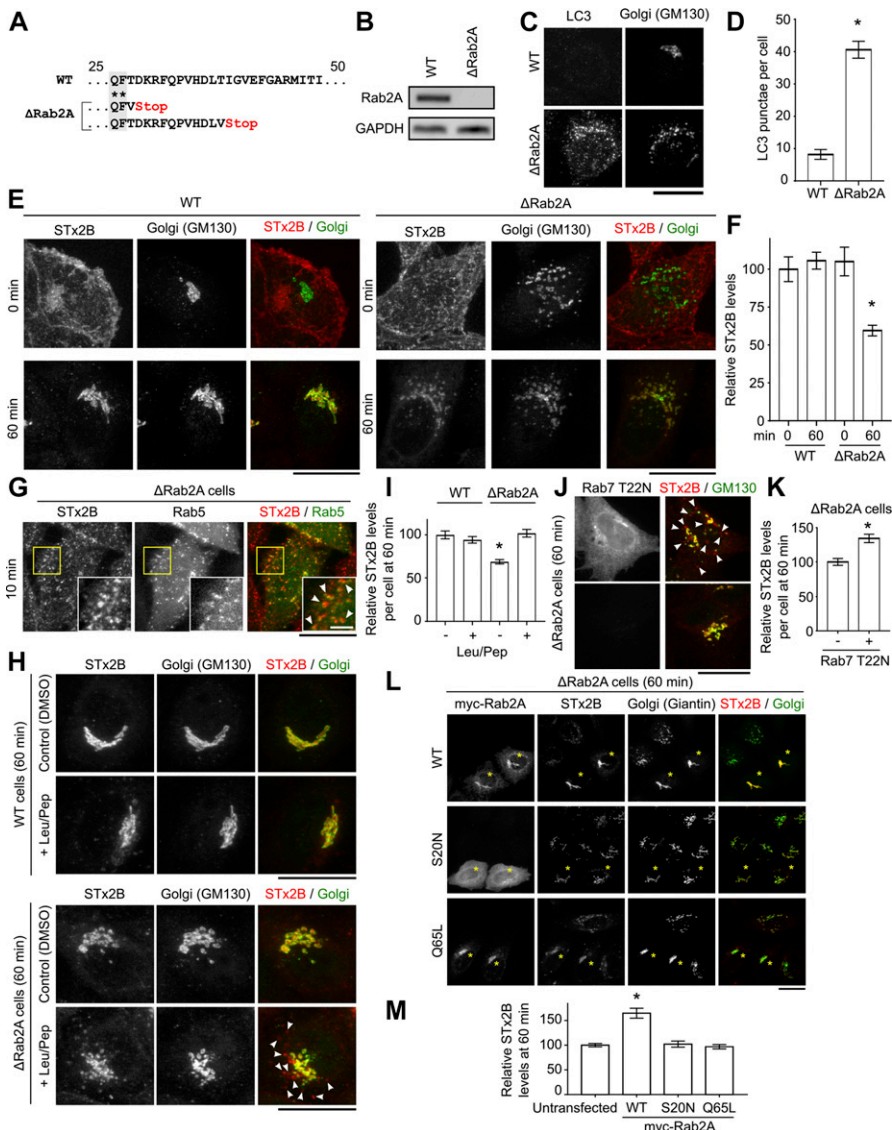

**Figure 1. Rab2a is required for the early endosome-to-Golgi transport of STx2B.**
**(A)** Genomic DNA sequences. **(B)** RT–PCR. **(C)** Immunofluorescence. Scale bar, 25 $\mu$m. **(D)** Quantification from (C). N = 15 cells per condition. *$P <$ 0.05 by $t$ test. **(E)** STx2B transport. Scale bars, 25 $\mu$m. **(F)** Quantification from (E). WT 0 min normalized to 100. N > 15 cells per condition. *$P <$ 0.05 by one-way ANOVA and Dunnett's post hoc test for comparison between WT at 0 min and other groups. **(G)** STx2B transport 24 h after transfection with Rab5$_{WT}$. Arrows denote overlap of STx2B with Rab5. Scale bars, 25 $\mu$m; inset 5 $\mu$m. **(H)** STx2B transport with or without exposure to leupeptin (leu) and pepstatin (pep) for 24 h. Arrows show STx2B signal in cytoplasmic punctae. Scale bars, 25 $\mu$m. **(I)** Quantification from (H). Levels in WT cells not exposed to leupeptin/pepstatin normalized to 100. N ≥ 25 cells per condition. *$P <$ 0.05 by one-way ANOVA and Dunnett's post hoc test for comparison between WT cells not exposed to leupeptin/pepstatin and other conditions. **(J)** STx2B transport 24 h after transfection with dominant negative Rab7 (Rab7$_{T22N}$). Arrows denote STx2B signal in cytoplasmic punctae. Scale bars, 25 $\mu$m. **(K)** Relative STx2B levels 24 h posttransfection. Levels in untransfected cells normalized to 100. N > 15 cells per condition. *$P <$ 0.05 by $t$ test. **(L)** Transport of STx2B 24 h posttransfection. Asterisks indicate transfected cells. Scale bar, 25 $\mu$m. **(M)** Quantification from (L). Values in untransfected cells normalized to 100. N > 20 cells per condition. *$P <$ 0.05 by one-way ANOVA and Dunnett's post hoc test for comparison between untransfected and other groups.

autophagosomes requires ATG7 (36), and fusion of autophagosomes, but not late endosomes, with lysosomes requires syntaxin17 (35).

To test for the role of autophagy, we generated *ΔATG7* or *Δsyntaxin17* cells using CRISPR/Cas9. For both genes, the CRISPR/Cas9 system introduced stop codons in the genomic DNA and depleted transcript levels (Fig 2A–D). We used the mRFP-GFP-LC3 tandem reporter to assay for autophagosome formation and autophagosome–lysosome fusion. The tandem reporter fluoresces in the red and green channels when recruited to autophagosomes but fluoresces only in the red channel after autophagosome–lysosome fusion due to quenching of GFP fluorescence (33, 35). In *ΔATG7* cells, recruitment of the tandem reporter to punctate structures was inhibited under physiological or starvation conditions (Fig S3A and B), indicating that autophagosome formation was blocked. In *Δsyntaxin17* cells, recruitment of the tandem reporter to punctate structures was not blocked, but the relative decrease in

GFP-positive punctae observed in WT cells when autophagy was induced by starvation was not evident (Fig S3C and D), indicating that the fusion of autophagosomes with lysosomes was inhibited. Consistent with a block in autophagosome–lysosome fusion in *Δsyntaxin17* cells, levels of endogenous LC3 were also elevated (Fig S3E and F). Notably, however, positioning of Lamp2-positive lysosomes, which is indicative of lysosomal dysfunction (37), was unaffected in *Δsyntaxin17* or *ΔATG7* cells (Fig S3G and H). Thus, loss of ATG7 or syntaxin17 inhibited autophagy without affecting lysosomes. Importantly, transport of STx2B to the Golgi was not inhibited in *ΔATG7* or *Δsyntaxin17* cells (Fig 2E–J). Identical results were obtained when ATG7 or syntaxin17 was depleted using siRNA (Fig 2K–P). Overall, ATG7 and syntaxin17, and by extension autophagy, are not required for the early endosome-to-Golgi transport of STx2B.

Subsequently, we depleted Vps39 using siRNA (we could not generate *ΔVps39* cells likely because knockout of Vps39 is lethal

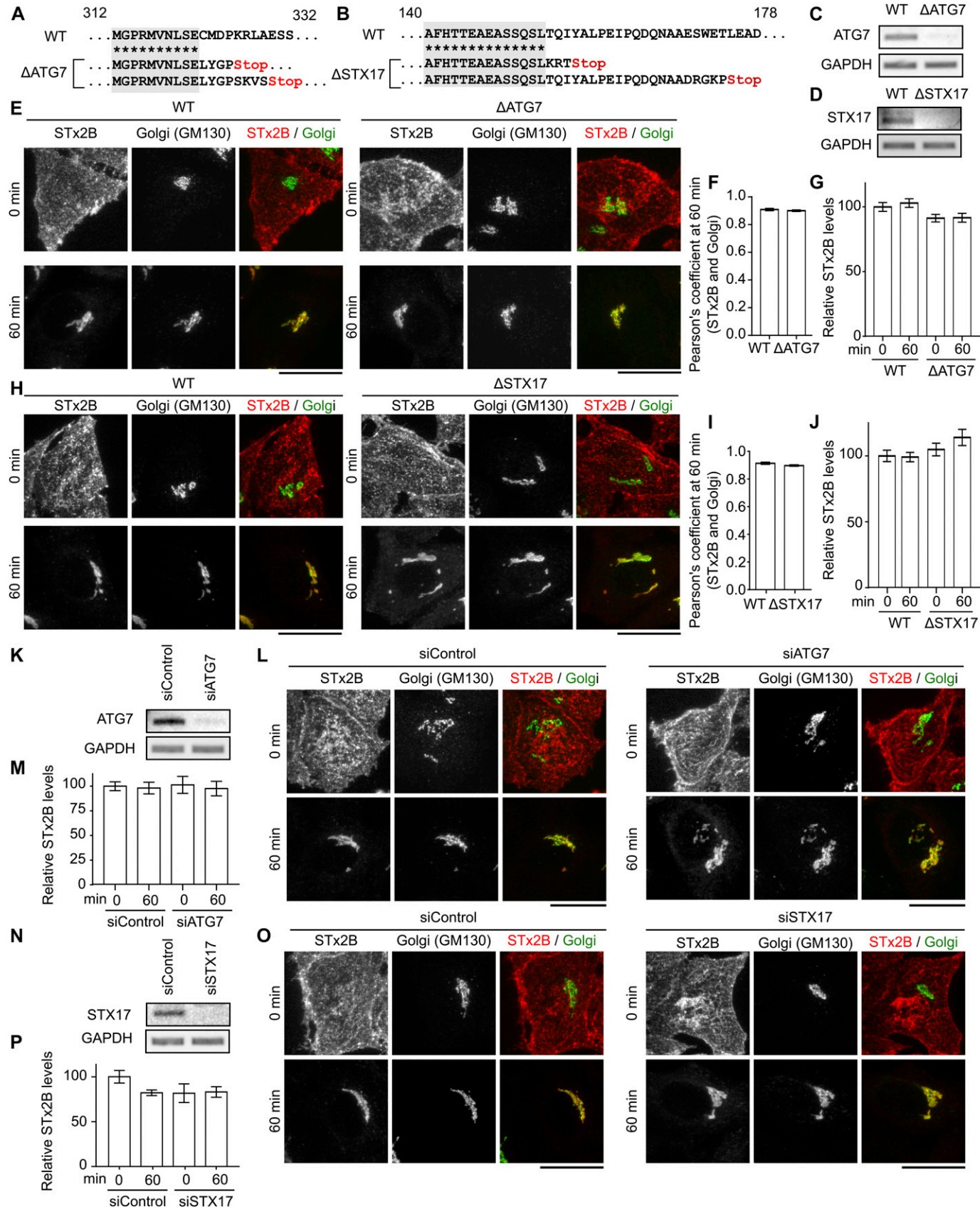

(38)). In knockdown cells, Vps39 transcript levels were depleted, endogenous LC3-positive punctae were enhanced, and Lamp2-positive lysosomes were clustered in the perinuclear area (Fig 3A–E), implying that lysosomal function and autophagy were compromised. In Vps39-depleted cells, STx2B bound the cell surface and trafficked to Rab5-positive early endosomes, but then failed to traffic to the Golgi, and was degraded (Fig 3F–H). Under these conditions, STx2B was detected in Rab7-positive late endosomes (Fig 3I), suggesting that failure to undergo early endosome-to-Golgi transport likely induced toxin degradation in late endosomes. The block in trafficking was rescued by expression of siRNA-resistant Vps39 (Fig 3J–L). The requirement of Vps39, but not ATG7 or syntaxin17, for STx2B trafficking implies that toxin transport depends on the function/biogenesis of lysosomes but not on the autophagy pathway.

### Inhibition of late endosome–lysosome fusion alters recruitment of retromer to early endosomes

By blocking late endosome–lysosome fusion, loss of Vps39 inhibits the protein degradation function of lysosomes as well as biogenesis of mature lysosomes (34). Treatment with lysosomal protease inhibitors did not block STx2B trafficking (Fig 3M and N), implying that toxin transport was independent of degradative function per se. Notably, in Vps39-depleted cells, endosomal maturation is also inhibited (39), likely due to defects in late endosome–lysosome fusion. Further, membrane recruitment of retromer, which is required for the early endosome-to-Golgi transport of STx2B (9), is linked to endosomal maturation (40). Thus, the block in late endosome–lysosome fusion in Vps39-depleted cells could indirectly inhibit early endosome-to-Golgi transport of STx2B by altering endosomal maturation and retromer function. Consistent with this, in Vps39 knockdown cells, EEA1- or SNX1-positive early endosomes were clustered in the perinuclear region, and association of the retromer component Vps26 with endosomal membranes was enhanced (Fig 3O–U). Since retromer function depends on its cyclic association with and dissociation from endosomal membranes, the increased association of Vps26 with early endosomes observed in Vps39-depleted cells may contribute to the block in STx2B transport. In totality, fusion of late endosomes, but not autophagosomes, with lysosomes is required for the early endosome-to-Golgi trafficking of STx2B, while function of lysosomes and the autophagy pathway is dispensable.

### TAM is a potent inhibitor of STx1 and STx2 transport and toxicity

It is challenging to bring a new small molecule into therapeutic use. An alternative approach is to repurpose a drug approved for treatment of another disease. Small molecules that increase the

pH of the endolysosomal compartment inhibit fusion of late endosomes with lysosomes (41, 42), block early endosome-to-Golgi protein transport (43), and protect against STx1-induced cytotoxicity (44). Notably, our prior studies indicate that there are critical differences in the molecular factors required for the trafficking of STx1 and STx2 (7,8). Therefore, we could not predict whether alterations of endolysosomal pH would effectively inhibit trafficking and toxicity of STx2. However, as several drugs currently approved by the Food and Drug Administration alter pH of the endolysosomal compartment, based on the totality of the above observations, it was reasonable to test whether one of these approved drugs could be repurposed as a STx2 transport inhibitor.

We validated that treatment with the V-ATPase inhibitor bafilomycin A1 (BFA1) (41) robustly inhibited the transport of STx2B to the Golgi apparatus (Fig 4A and B). A subsequent screen of Food and Drug Administration–approved lysosome-targeting drugs identified two compounds that increase endolysosomal pH, TAM and chloroquine (CLQ) (41, 45, 46, 47), to be inhibitors of STx2B transport (Fig 4A and B). CLQ and TAM are lysosomotropic weak bases that accumulate within lysosomes/acidic compartments and directly increase pH (41, 45, 46, 47) (see below). Subsequent studies focused on TAM, which had a greater inhibitory effect and is currently approved for breast cancer therapy (48, 49). Similar to Vps39-depleted cells, in TAM-treated cells, LC3-positive punctae were elevated, and while STx2B bound the cell surface and reached Rab5-positive early endosomes, the toxin failed to traffic to the Golgi and was degraded (Fig 4A–C). STx2B was detected in Rab7-positive late endosomes after TAM treatment (Fig 4D); as with Vps39 depletion, blocked early endosome-to-Golgi transport likely induced transit of the toxin to degradative late endosomes. Note that apparent differences in control panels for Rab7/STx2B in Figs 4D and 3I may be reflective of different treatment conditions and/or expression level of the transfected Rab7 construct. TAM also inhibited the transport of STx1 B-subunit (STx1B) to the Golgi and induced degradation of STx1B (Fig 4E and F).

### TAM protects cells against STx1 and STx2 toxicity by acting as a weak base that directly increases endolysosomal pH

Our next goals were to determine whether TAM could protect cells against STx1- or STx2-induced death and elucidate its mechanism of action. Treatment with 10 $\mu$M TAM provided ~200-fold protection against STx2-induced cell death and ~50-fold protection against STx1 without inducing cytotoxicity (Fig 5A). Protection was evident at TAM doses as low as 2.5 $\mu$M (Fig 5B).

We hypothesized that the protective effect of TAM would be related to its capability to increase endolysosomal pH. Presence of a tertiary amine makes TAM a weak base (Fig 5C) (46). Prior studies indicate that this weak base property allows TAM to directly titrate

---

**Figure 2. The autophagy pathway is not required for STx2B trafficking.**
**(A, B)** Genomic DNA sequences. STX17, syntaxin17. **(C, D, K, N)** RT–PCR. **(E, H, L, O)** STx2B transport imaged at 0 or 60 min. Scale bars, 25 $\mu$m. **(F, I)** Pearson's coefficient for colocalization between STx2B and the Golgi apparatus at 60 min from (E) and (H). N = 15 cells per condition. There were no differences between groups using t test. **(G, J, M, P)** STx2B levels from (E, H, L, O). Levels at 0 min in WT cells (G, J) or cells transfected with control siRNA (M, P) normalized to 100. N > 15 cells per condition. There were no differences between WT or control siRNA-transfected cells at 0 min and other groups using one-way ANOVA and Dunnett's post hoc test.

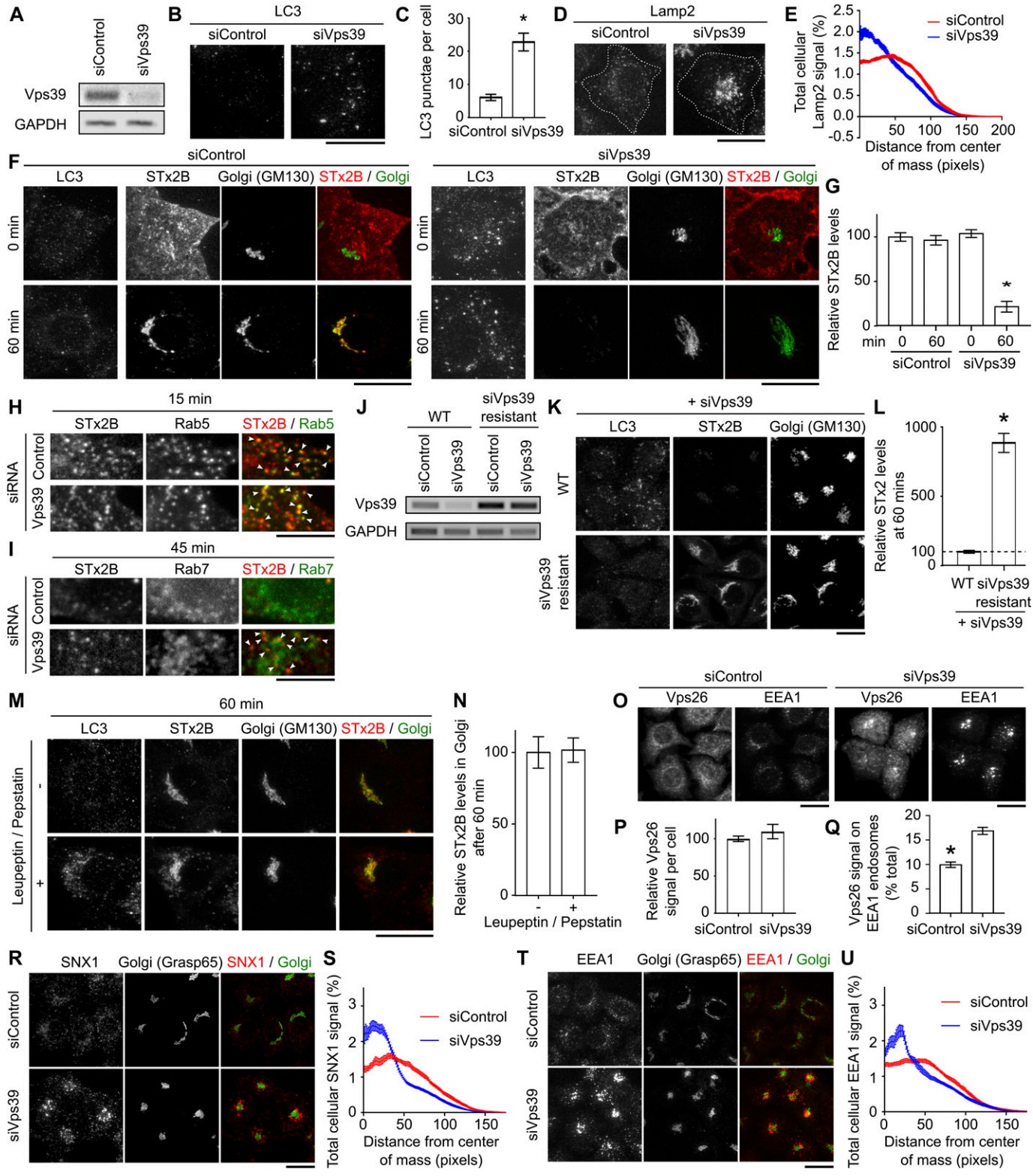

**Figure 3. Depletion of Vps39 blocks the early endosome-to-Golgi transport of STx2B.**
**(A)** RT–PCR. **(B, D)** Immunofluorescence to detect LC3 or Lamp2. Scale bars, 25 $\mu$m. **(C, E)** Quantification of data from (B) and (D). N ≥ 15 cells per condition. *$P$ < 0.05 by $t$ test. **(F)** STx2B transport assay. Scale bars, 25 $\mu$m. **(G)** STx2B levels from (F). Levels in control-transfected cells at 0 min normalized to 100. N = 15 cells per condition. *$P$ < 0.05 by one-way ANOVA and Dunnett's post hoc test for the comparison between control 0 min and other groups. **(H, I)** STx2B transport in cells transfected with control or Vps39 siRNA. Cells were also transfected with plasmids encoding Rab5$_{WT}$ or Rab7$_{WT}$ 24 h prior to the transport assay. Arrows denote overlap of STx2B with Rab proteins. Scale bars, 10 $\mu$m. **(J)** RT–PCR in WT cells or cells stably overexpressing siRNA-resistant Vps39 after treatment with control or Vps39 siRNA. **(K)** STx2B transport at

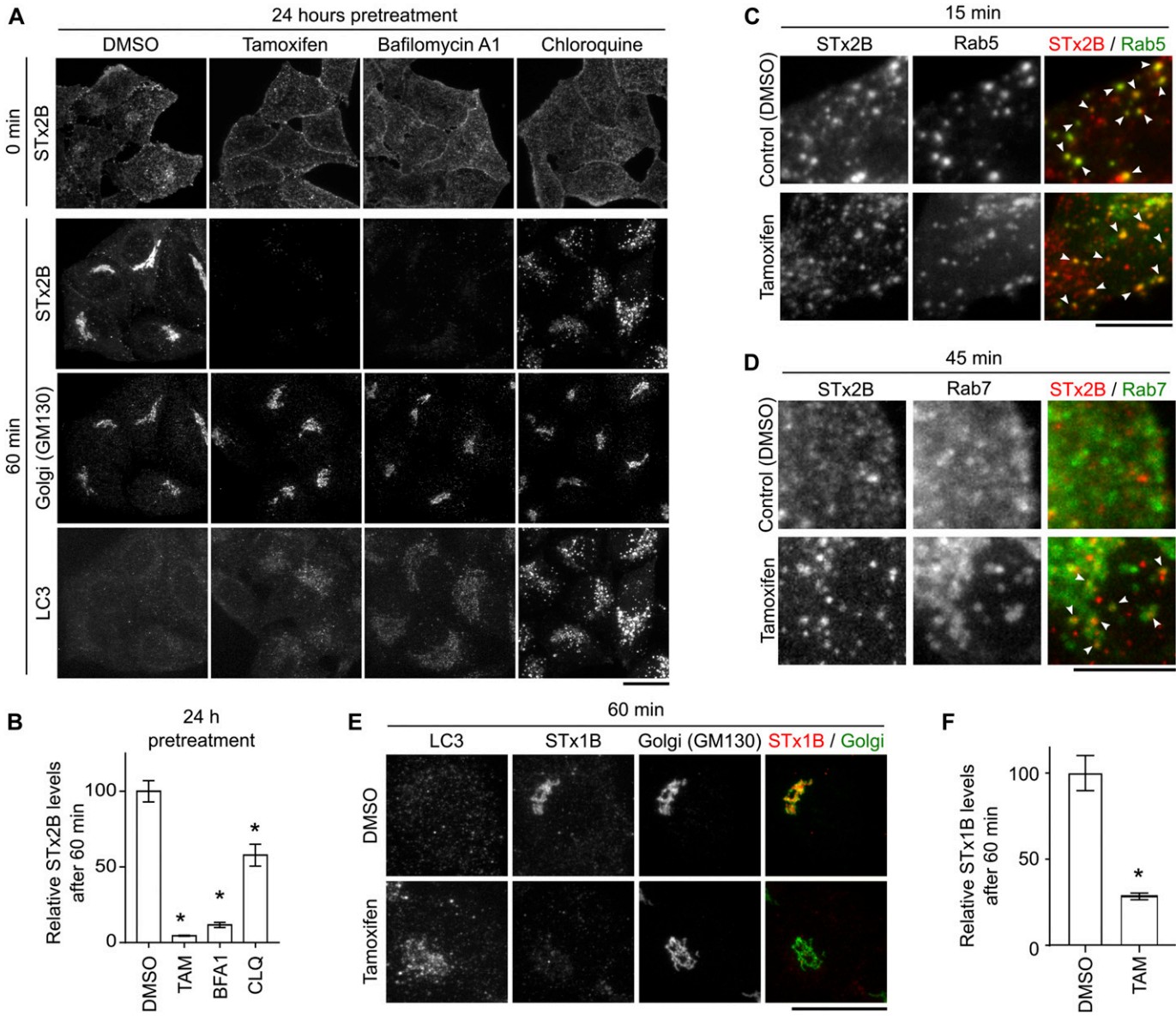

**Figure 4.   TAM inhibits retrograde trafficking of STx2B and STx1B.**
**(A)** STx2B transport in cells treated with DMSO (0.1%), TAM (10 $\mu$M), BFA1 (100 nM), or CLQ (50 $\mu$M) for 24 h. Scale bar, 25 $\mu$m. **(B)** Quantification of data from (A) with levels in DMSO-exposed cells at 60 min normalized to 100. N = 25 cells per condition. *$P < 0.05$ by one-way ANOVA and Dunnett's post hoc test for comparison between DMSO and other groups. **(C, D)** STx2B transport in cells transfected with Rab5$_{WT}$ or Rab7$_{WT}$ for 24 h and subsequently treated with DMSO or TAM for an additional 24 h. Arrows denote overlap of STx2B with Rab proteins. Scale bars, 10 $\mu$m. **(E)** STx1B transport in cells treated with DMSO or 10 $\mu$M TAM for 24 h. Scale bar, 25 $\mu$m. **(F)** Quantification from (E) as described for (B). N > 15 cells. *$P < 0.05$ by t test.

the pH of endolysosomal compartments upward (i.e., increase endolysosomal pH), and that TAM-mediated changes in endolysosomal pH are independent of estrogen receptors or any cellular protein (45, 46). HeLa cells do not express estrogen receptors (50), ruling out the role of estrogen signaling in our assays. If the protective effect of TAM was based on an increase in endolysosomal pH, compounds that lack the tertiary amine and cannot function as a weak base should not protect against STx2 toxicity. Consistent with this prediction, three clinically approved compounds with the tertiary amine, toremifene (TOR), raloxifene (RAL),

60 min in WT cells or cells stably overexpressing siRNA-resistant Vps39 after treatment with Vps39 siRNA. Scale bar, 25 $\mu$m. **(L)** STx2B levels from (K). Levels in WT cells normalized to 100. N ≥ 30 cells per condition. *$P < 0.05$ by t test. Scale bar, 25 $\mu$m. **(M)** STx2B transport assays in cells treated with or without leupeptin and pepstatin for 24 h. Scale bar, 25 $\mu$m. **(N)** Quantification of the relative amounts of STx2B in the Golgi apparatus from (M) with values in cultures not exposed to leupeptin/pepstatin normalized to 100. N = 15 cells per condition. **(O, R, T)** Immunofluorescence. Scale bar, 25 $\mu$m. **(P, Q, S, U)** Quantification of data from (O, R, T). N = 15 cells per condition. *$P < 0.05$ by t test.

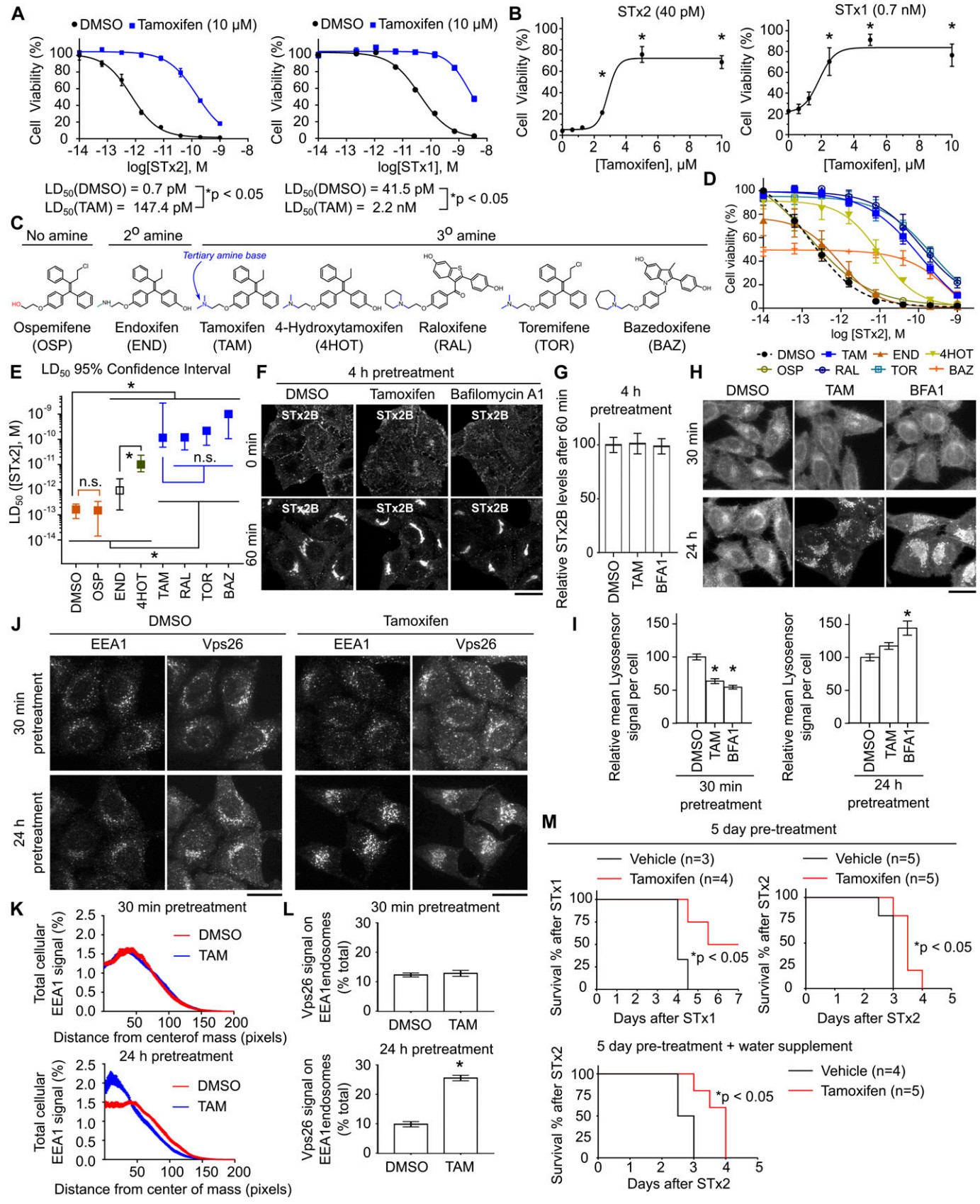

and bazedoxifene (BAZ), provided as much protection as TAM against STx2-induced cell death (Fig 5C–E). TOR and RAL did not alter cell viability by themselves, but BAZ had noticeable cyto-toxicity at concentrations used (Fig 5D). A lower level of protection was evident with the metabolite 4-hydroxytamoxifen (4HOT), which also has the tertiary amine (Fig 5C–E); the reduced protective effect was likely due to the presence of the aliphatic hydroxyl group that may inhibit membrane incorporation. In contrast, protection pro-vided by the metabolite endoxifen (END), which has a secondary amine and is a weaker base than the tertiary amine–containing compounds, was substantially weaker than TAM or 4HOT (Fig 5C–E). Furthermore, ospemifene (OSP), which does not have an amine group and is not a weak base, failed to protect all together (Fig 5C–E). We verified that TOR blocked transport of STx2B to the Golgi, but OSP did not (Fig S4A and B). Thus, the tertiary amine group of TAM is necessary to protect against toxin-induced cell death, im-plying that protection is provided by the weak base effect of TAM.

To directly determine the mechanism by which increased endolysosomal pH impacted toxin transport and toxicity, we per-formed time course assays. A 24 h pretreatment with TAM or BFA1 was necessary to block STx2B transport, and treatment for shorter durations (e.g., 4 h) did not have an effect (Figs 5F and G, and 4A and B). However, similar to several lysosomotropic compounds, TAM induces a transient change in endolysosomal pH with an increase evident at early time points (30–60 min) but not 24 h after treatment (47). We verified these results for TAM and observed a similar effect with BFA1 (Fig 5H and I). A possibility is that a change in endoly-sosomal pH initiates a cascade of events that leads to a block in transport at a later time point. Consistent with this, morphological differences were apparent in the endolysosomal compartments of cells treated with TAM or BFA1 for 24 h (Fig 5H). Furthermore, similar to results obtained with Vps39 depletion, 24 h, but not 30 min, after TAM treatment, positioning of early endosomes gained a peri-nuclear prominence and recruitment of the retromer component Vps26 to early endosomes was enhanced (Fig 5J–L). Additionally, TOR, which has the tertiary amine group, increased endolysosomal pH at 30 min and increased Vps26 levels on early endosomes at 24 h, while OSP, which lacks the tertiary amine, did not (Fig S4C–F). Overall, our results indicate that TAM phenocopies Vps39 depletion with regards to protein transport at the early endosome/Golgi interface and inhibits transport and toxicity of STx1 and STx2.

### TAM protects mice against lethal STx1 or STx2 toxicosis

To test the disease relevance of our results, we performed ex-periments at the whole organism level using a mouse model in which animals were given a single intraperitoneal injection of STx1 or STx2. This model produces fulminant toxicosis and the observed renal damage has similarities with changes seen in human patients (6, 12, 51). We pretreated the animals with 70 mg TAM/kg body weight or vehicle for 5 d before toxin exposure. The pretreatment regimen was utilized because, in humans, the toxins enter the bloodstream ~10–14 d after bacterial infection and ~4–7 d after development of symptomatic diarrhea (2, 3), providing a window of opportunity for diagnosis and initiation of antitoxin therapy. Additionally, mice are more efficient at handling TAM than humans, and our regimen was designed to produce serum levels similar to humans receiving TAM at the currently approved dose of 20 mg/d (48, 49). Vehicle-treated mice exposed to 50 ng STx1/g body weight died within 3–4 d (Fig 5M). As expected, STx2 was more toxic, and vehicle-treated mice ex-posed to 2.8 ng STx2/g body weight died within 2–3 d (Fig 5M). Importantly, TAM treatment significantly improved the survival of STx1- or STx2-treated mice (Fig 5M). In the STx1 group, a subset of TAM-treated mice remained healthy until the end of the experiment (Fig 5M). TAM-treated mice that received STx2 also survived for significantly longer than those that received vehicle (Fig 5M). A similar level of protection against STx2 was evident when, in ad-dition to the pretreatment, TAM was also orally delivered via drinking water after STx2 injection (Fig 5M).

## Discussion

The early endosome-to-Golgi transport of STx2 was dependent on efficient late endosome–lysosome fusion. The role of late endosome–lysosome fusion in early endosome-to-Golgi transport has not received much attention. However, a strong effect could be predicted because inhibition of late endosome–lysosome fusion is expected to block endosomal maturation, and recruitment of retromer to endosomal membranes depends on endosomal maturation (40). Indeed, direct experimental evidence for a block in endosomal maturation on depletion of Vps39 is available (39), and in our studies, loss of Vps39 impacted endosomal recruitment of retromer. Clearly, it will be important to better understand the mechanisms by which association of retromer with endosomal membranes is modulated by late endosome–lysosome fusion in the future. There are several additional issues worth noting in this discussion. Retromer is a master regulator of cargo export from endosomes (40). Therefore, late endosome–lysosome fusion may have a broad effect on cargo trafficking at the early endosome/Golgi interface. By extension, phenotypic presentations of human

---

**Figure 5. TAM acts as a weak base to protect cells and mice against lethal STx1 or STx2 toxicosis.**
**(A)** Viability assays in cells treated with vehicle or 10 $\mu$M TAM for 24 h followed by exposure to indicated concentrations of STx1 or STx2 for 16 h in the presence of vehicle or TAM. N = 3. *$P$ < 0.05 by nonlinear regression. **(B)** Viability as in (A) using varying concentrations of TAM and 40 pM STx2 or 0.7 nM STx1. N ≥ 3. *$P$ < 0.05 by one-way ANOVA and Dunnett's post hoc test for comparison between no TAM and other conditions. **(C)** Chemical structures. **(D, E)** Viability assays as in (A) using 10 $\mu$M of each compound and indicated concentrations of STx2. LD$_{50}$ with 95% confidence interval depicted in (E). N ≥ 3 per compound. *$P$ < 0.05 by nonlinear regression. **(F)** STx2B transport after treatment with indicated compounds for 4 h. Scale bars, 25 $\mu$m. **(G)** Quantification of data from (F) with levels in DMSO-exposed cells at 60 min normalized to 100. N > 15 cells per condition. There were no differences between groups using one-way ANOVA. **(H)** LysoSensor signal in cells treated with indicated compounds for 30 min or 24 h. Scale bars, 25 $\mu$m. **(I)** Quantification of mean Lysosensor signal per cell from (H). N ≥ 15 cells per condition. *$P$ < 0.05 by one-way ANOVA and Dunnett's post hoc test for comparison between DMSO and other groups. **(J)** Immunofluorescence to detect EEA1 and Vps26 in cultures exposed to DMSO or 10 $\mu$M TAM for 30 min or 24 h. Scale bars, 25 $\mu$m. **(K, L)** Quantification of data from (J). N = 15 cells per condition. *$P$ < 0.05 by $t$ test. **(M)** Mouse survival by the method of Kaplan–Meier. *$P$ < 0.05 by the Gehan–Breslow–Wilcoxon and log-rank (Mantel–Cox) tests. N.S., not significant.

diseases linked to defects in lysosomal biogenesis, such as Dannon disease due to mutations in *LAMP2* (52), may be influenced by indirect effects on early endosome-to-Golgi transport of endogenous cargo proteins. Furthermore, transport of related exogenous toxins that traffic to the Golgi from endosomes (e.g., cholera) (5) may be similarly influenced by late endosome–lysosome fusion. Finally, unlike GPP130 for STx1B (6, 7), a transmembrane endosomal receptor has not been identified for STx2B (or related toxins). If such a STx2B receptor exists, it may accumulate in the endosomal compartment of cell lines with deficits in retromer function or late endosome–lysosome fusion. Proteomic analyses of endosomal fractions generated from such cell lines may identify this putative receptor and provide needed breakthroughs in understanding how cargo proteins that are restricted to the endosomal lumen communicate with cytosolic trafficking factors, such as retromer, to sort out endosomes and traffic to the Golgi.

A major goal of our study was to translate our mechanistic work into a therapeutically viable strategy to block STx2 transport. Small molecules that increase endolysosomal pH were reported to exert an inhibitory effect on the fusion of late endosomes with lysosomes (41, 42), block protein transport from early endosomes to the Golgi (43), and confer protection against STx1-induced cytotoxicity (44). Therefore, we screened for the effects of clinically approved drugs that alter endolysosomal pH on STx2 transport and toxicity. We identified TAM to be a potent inhibitor of the early endosome-to-Golgi transport and cytotoxicity of STx2 as well as STx1 in cells. While TAM is a well-known selective estrogen receptor modulator used for breast cancer therapy for its antiestrogenic properties (48, 49), it is also a lysosomotropic compound that directly increases endolysosomal pH by acting as a weak base (45, 46, 47). The protective effect of TAM against STx2 and STx1 was mediated by its weak base property and was independent of estrogen receptors. Indeed, three compounds that retain the weak base activity of TAM (TOR, RAL, and bazodoxifene) also protected against STx2-induced cell death. Notably, similar to Vps39 depletion, TAM-mediated changes in endolysosomal pH altered recruitment of retromer to early endosomal membranes. The phenotypic similarities were not surprising because, as described earlier, late endosome–lysosome fusion is inhibited by Vps39 depletion (33, 34, 35) or increased endolysosomal pH (41, 42), deficits in late endosome–lysosome fusion are expected to indirectly alter endosomal maturation, and endosomal recruitment of retromer depends on endosomal maturation (40).

Treatment with TAM enhanced the survival of mice injected with a lethal dose of STx1 or STx2. Two reasons may explain why a greater level of protection was not seen in TAM-treated mice. First, the intraperitoneal injection model induces fulminant toxicosis, unlike that produced by bacterial infection in human patients where lower amounts of the toxins enter circulation over a longer period. Further, unlike rats or humans, mice rapidly metabolize TAM to 4HOT (49), which we discovered was not as effective as TAM in protecting against STx2-induced cell death. While the observed protection in mice using a severe toxicosis model suggests that TAM may be effective for the treatment of human patients infected with STEC bacteria, additional in vivo experiments using a bacterial infection model are necessary before human therapy can be contemplated. TOR and RAL, which like TAM, robustly protected cells against STx2-induced death without inducing cytotoxicity may also be therapeutically useful. Unlike other drugs in development, as TAM, TOR, and RAL are already approved for use in humans, this line of work holds the promise of rapid clinical translation.

# Materials and Methods

## Experimental design

Assays in cell culture were designed such that differences between control and experimental (i.e., knockout, dicer or siRNA-depleted, or drug-treated) groups could be compared using standard statistical tests described in the Statistical analyses section below and in individual figure legends. All cell culture experiments were replicated at least three times independently. Animal experiments were performed using vehicle- or TAM-treated mice, and differences between groups were compared statistically using methods described in the Statistical analyses section.

## Cell culture and generation of knockout and overexpression clones

WT cells were a HeLa cell line that stably overexpressed globotriaosylceramide, the cell surface receptor for STx1B and STx2B. We have used this subline for numerous assays on STx1 and STx2 over the last few years (8, 9). Culture conditions were identical to those described previously (8, 9). Mutations in genomic DNA were introduced using a lentivirus-based CRISPR/Cas9 system described by us recently (8). The guide RNA sequences were as follows: Rab2a: 5′-CCA GTG CAT GAC CTT ACT AT-3′; ATG7: 5′-GGT GAA CCT CAG TGA ATG TA-3′; and syntaxin17: 5′-ATC AAA ATG CTG CAG AAT CG-3′. Other procedures (production of lentivirus, infection of target cells with lentivirus, selection of single cell clones, and sequencing of genomic DNA) were exactly as described in detail in our reference 8 and therefore are not described here. To generate cells that overexpressed siRNA-resistant Vps39, WT cells were infected with lentivirus, in which the transfer plasmid encoded Vps39 in a pLJM1 backbone vector (plasmid #34611; Addgene), using procedures identical to those described by us previously (8). The sequence of Vps39 in the transfer plasmid had seven silent mutations in the region targeted by the siRNA (5′-CAA CCA CCA TAT ATA ATC GCT-3′) so that the overexpressed construct was resistant to siRNA-mediated depletion.

## Transient transfections using Dicer-substrate short interfering RNAs, siRNA, and plasmid DNA

Dicer-substrate short interfering RNAs targeting FUT1 or STAM and the control RNA, which did not target any human gene, were obtained from Integrated DNA Technologies (FUT1: hs.Ri.FUT1.13.3; STAM: hs.Ri.STAM.13.3; Control: #51-01-14-03). For transfections, cells were grown to ~40% confluency and transfected with 10 nM final concentration of each RNA duplex using Oligofectamine transfection reagent (Invitrogen) following manufacturer's instructions. Cultures were used for experiments 48 h after transfection.

Transfections with siRNAs were performed using Oligofectamine reagent essentially as described by us previously (8, 9). Sequences of GBF1 and control siRNAs were described by us previously (8, 53). Sequences of other siRNAs used were as follows: Vps39 sense: 5′-GCCUCCCUACAUCAUUGCATT-3′, antisense: 5′-UGCAAUGAUGUAGGG-AGGCTG-3′; ATG7 sense: 5′-GCCAGAGGAUUCAACAUGATT-3′, antisense: 5′-UCAUGUUGAAUCCUCUGGCTT-3′; and syntaxin17 sense: 5′-GGAAACCUUAGAAGCGGACUUAAUU-3′, antisense: 5′-AAUUAAGUCC-GCUUCUAAGGUUUCC-3′. Except Vps39, experiments were performed 72 h posttransfection. To obtain robust Vps39 knockdown, it was necessary to transfect each culture with siRNA two times. The second transfection was performed 48 h after the first, and cultures were analyzed 5 d after the first transfection. A similar protocol was used in prior studies to deplete Vps39 using siRNA (39).

Plasmid DNA was transfected using JetPEI reagent (VWR) as described by us previously (8, 9). Constructs encoding GFP-Rab5$_{WT}$, GFP-Rab7$_{WT}$, and GFP-Rab7$_{T22N}$ (dominant negative) have been previously described by us (8, 9). Plasmid encoding myc-tagged human Rab2a was from Addgene (plasmid #46779). Mutations were introduced into this plasmid using QuikChange (Agilent Technologies) (9).

### PCR and RT–PCR

PCR and RT–PCR were performed as described previously (8). Primers used for RT–PCR were as follows: Rab2a FWD: 5′-CAG ACA AGA GGT TTC AGC CAG TGC-3′; Rab2a REV: 5′-GCT CCT GCT GCA CCT CTG TAA TAC-3′; FUT1 FWD: 5′-GCC CTG CTC ACA CAG TGC AAC C-3′; FUT1 REV: 5′-GGC TTA GCC AAT GTC CCA GAG TGG-3′; STAM FWD: 5′-CTC TCA GCC AGG CAG TGG TCC-3′; STAM REV: GCA GTA GCG GCA GGA GGA GG-3′; ATG7 FWD: 5′-AGT GAC GAT CGG ATG AAT GA-3′; ATG7 REV: 5′-TGG TCT CAT CAT CGC TCA TGT-3′; syntaxin17 FWD: 5′-TCG TGG GAA ACCT TA GAA GCGG-3′; syntaxin17 REV: 5′-GCA GCA CTG TTG ACA TGG TCT Gg-3′; Vps39 FWD: 5′-CCT GAA CTG GAC GGA CAT ACC A-3′; Vps39 REV: 5′-CTT TGG ACC AGA AGC CTC GGT T-3′; GAPDH FWD: 5′-GGC TAC ACT GAG CAC CAG GTG-3′; and GAPDH REV: 5′-GGT CCA CCA CCC TGT TGC TGg-3′.

### Antibodies

Sources of antibodies used were as follows: monoclonal anti-GM130 (#610822), anti-EEA1 (#610456), and anti-SNX1 (#611482) from BD Biosciences; monoclonal anti-Lamp2 (ab25631) and polyclonal anti-Vps26 (ab23892) from Abcam; and polyclonal anti-LC3 A/B (D3U4C) from Cell Signaling Technologies. Polyclonal anti-giantin and anti-GRASP65 antibodies were described previously (8, 9).

### STx1B and STx2B transport assays

Transport assays using fluorescently labeled untagged STx1B or His-tagged STx2B were performed exactly as described by us recently (8, 9). Briefly, cells were washed with ice-cold phosphate-buffered saline (three times). After this, cells were incubated with 2 µg/ml of STx2B or 5 µg/ml of STx1B in transport media (Dulbecco's modified Eagle's medium supplemented with 10% fetal bovine serum, 100 IU/ml penicillin-G and 100 µg/ml streptomycin) for 30 min on ice at 4°C to allow binding of toxin to the cell surface. Cells

were then again washed with ice-cold phosphate-buffered saline (three times) and transferred to toxin-free transport media at 37°C to initiate toxin transport. Cultures were fixed after start of transport at times indicated in each figure and processed for microscopy.

### Drug treatments in cell culture and viability assays

TAM, TOR, RAL, BAZ, 4HOT, END, OSP, BFA1, and CLQ were purchased from Sigma-Aldrich. TAM was used at 10 µM unless specified otherwise. TOR, RAL, BAZ, 4HOT, END, and OSP were all used at 10 µM. BFA1 was used at 100 nM, and CLQ was used at 50 µM. DMSO was added at 0.1% when used as a vehicle control. Leupeptin and pepstatin were used at final concentrations of 100 and 50 µg/ml, respectively, as described by us previously (8, 9). Compounds were present in the media during transport assays performed using STx1B or STx2B and during exposure to STx1 or STx2 holotoxins, which were obtained from BEI Resources. Cell viability was assessed using 3-(4,5-dimethylthiazol-2-yl)-2,5-diphenyltetrazolium bromide reagent, as described by us recently (9).

### Microscopy and image analyses

Immunofluorescence staining was performed as described in our recent publications (8, 9). For assessing pH of endolysosomal compartments, LysoSensor Green DND-189 probe (Thermo Fisher Scientific) was used at 1 µM. Cells were exposed to the probe for 30 min, and live cultures were imaged immediately.

For imaging, a swept-field confocal microscope equipped with a four-line high-power laser launch and a 100× 1.45 N.A. oil immersion objective (Nikon) was used. The camera was an iXon3 X3 DU897 electron-multiplying charge-coupled device camera (Andor Technology). All images were captured as z-stacks with 0.2-µm spacing between individual frames. Images depicted in the figures are maximum-intensity projections of the stacks.

All analyses were performed using ImageJ (National Institutes of Health; http://rsb.info.nih.gov/ij/index.html). Particle counts were quantified using the *Analyze Particles* function; identical thresholds were used for control and experimental samples. Average fluorescence values per cell and Pearson's coefficient for colocalization were determined as described previously (8, 9). To quantify data obtained from the tandem mRFP-GFP-LC3 reporter, we quantified the percent of RFP-positive punctae that were also GFP-positive using the *ComDet* spots colocalization plugin. The Vps26 signal on EEA1-positive endosomes was measured as the percent of Vps26 signal in regions positive for EEA1 relative to the total cellular levels of Vps26 for each cell. EEA1 regions were identified for individual cells using the *ComDet* plugin. STx2B levels in the Golgi apparatus were quantified using the Golgi signal as the region of interest. To quantify perinuclear clustering of endosomal markers and lysosomes, the *Radial Profile* plugin was used on the average projection of acquired Z-stacks. Individual cells were outlined and isolated using the *Clear Outside* function. The center of mass of the measured signal was used as the radial center, and the distance distribution was measured over a 200 pixel (1,250 µm) radius.

## Mouse assays

All experiments with mice were approved by the Institutional Animal Care and Use Committee of UT Austin. 6–8-wk-old male Balb/c mice were used based on our prior work (6), and pilot studies showing that these animals develop lethal toxicosis when injected with STx1 or STx2. Animals received one daily intraperitoneal injection of TAM (70 mg TAM/kg body weight) in 100 μl sunflower oil (TAM group) or 100 μl sunflower oil (vehicle group) for 5 d. On the fifth day, animals received an additional intraperitoneal injection of STx1 (50 ng STx1/g body weight in 100 μl phosphate-buffered saline) or STx2 (2.8 ng STx1/g body weight in 100 μl phosphate-buffered saline). For animals that received oral TAM after toxin injection, TAM was provided in drinking water at an effective dose of 13 mg TAM/kg body weight/d while vehicle-treated animals received drinking water without TAM. After toxin injection, animals were monitored every 6 h for the onset of terminal morbidity at which point they were euthanized. Morbidly sick animals were positive for three of the following five signs: loss of >10% body weight, lethargy/decreased movement, dehydration, passage of loose stools, and onset of paralysis. Euthanasia was using carbon dioxide (54, 55).

## Statistical analyses

Statistical analyses were performed using GraphPad Prism 8 software (GraphPad). All cell culture experiments were independently replicated at least three times. $t$ test assuming equal variances was used to compare data between two groups. For comparisons between multiple groups, one-way ANOVA followed by Dunnett's or Tukey–Kramer post hoc test was used. Nonlinear regression was used to calculate the $LD_{50}$ of STx1 or STx2 in cell culture. Sample sizes for cell-based assays were based on power analyses and effect sizes and designed to detect differences between groups at 80% power with $P$ = 0.05. Animal survival was assayed using the method of Kaplan–Meier and the Gehan–Breslow–Wilcoxon and log-rank (Mantel–Cox) tests. Mouse sample sizes were based on previous studies by us and others that utilized similar numbers of animals in STx1/STx2 survival assays (6, 12). In all analyses, $P$ < 0.05 was considered statistically significant. Asterisks in graphs represent statistically significant differences.

# Supplementary Information

# Acknowledgements

Supported by the National Institutes of Health/National Institute of Allergy and Infectious Diseases grant R21-AI123608 and UT Austin start-up funds (S Mukhopadhyay).

## Author Contributions

AS Selyunin: conceptualization, data curation, formal analysis, investigation, methodology, and writing—original draft, review, and editing.

S Hutchens: investigation.

SF McHardy: conceptualization.

S Mukhopadhyay: conceptualization, data curation, formal analysis, supervision, funding acquisition, investigation, methodology, project administration, and writing—original draft, review, and editing.

## Conflict of Interest Statement

The University of Texas at Austin has filed a provisional patent application (inventors: AS Selyunin, SF McHardy, and S Mukhopadhyay) on the use of tamoxifen for treatment of STEC infections.

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
