## [Reviewer comments · Life Science Alliance]

Tamoxifen blocks retrograde trafficking of Shiga toxin 1 and 2 and protects against lethal toxicosis

Andrey S. Selyunin, Steven Hutchens, Stanton F. McHardy, and Somshuvra Mukhopadhyay
DOI: 10.26508/lsa.201900439

Review timeline:

Submission Date:	26 May 2019
Editorial Decision:	27 May 2019
Revision Received:	3 June 2019
Editorial Decision:	6 June 2019
Revision Received:	6 June 2019
Accepted:	7 June 2019

Report:

(Note: Letters and reports are not edited. The original formatting of letters and referee reports may not be reflected in this compilation.)

Please note that the manuscript was previously reviewed at another journal and the reports were taken into account in inviting a revision for publication at *Life Science Alliance* prior to submission to *Life Science Alliance*.

No Peer Review Process File is available with this article, as the authors have chosen not to make the review process public in this case.

Thank you for transferring your manuscript entitled "Tamoxifen blocks trafficking of Shiga toxin 1 and 2 and protects against lethal toxicosis" to Life Science Alliance. The manuscript was assessed by expert reviewers at another journal before, and the editors transferred those reports to us with your permission.

The reviewers found your results potentially interesting, but thought that your conclusions were not sufficiently supported by the data provided and they would have expected further reaching mechanistic insight and additional *in vivo* studies. The latter is not needed for publication in Life Science Alliance, and we would thus like to invite you to submit a revised version to us. The request for further *in vivo* studies should get addressed by acknowledging the need for this in the manuscript text. Further mechanistic insight into the re-routing aspects of the work is also not needed and the proposed involvement of the retromer should therefore be toned-down. However, better support for the conclusions that are based on some current datasets is needed:

- the Golgi is dispersed in the deltaRab2a cells and the support for a loss of co-localization needs to be strengthened. Repeating the experiments with Leu/Pep treatment may help to better illustrate the proposed re-routing. The conclusions regarding loss of co-localization in the siVps39 analysis similarly need to get better supported. (reviewer #1)
- reviewer #1's concerns regarding missing controls for Fig 3H, 3I, 4C, 4D need to get addressed
- tamoxifen effects on pH need to be shown (reviewer #1)
- reviewer #1's concerns (in minor comments) regarding number of cells analyzed for quantifications and missing quantifications need to get addressed

Thank you for this interesting contribution to Life Science Alliance. We are looking forward to receiving your revised manuscript.

Thank you for submitting your revised manuscript entitled "Tamoxifen blocks retrograde trafficking of Shiga toxin 1 and 2 and protects against lethal toxicosis". I now assessed your point-by-point response and the new data added to the manuscript.

I appreciate the text changes introduced to address the reviewer's criticisms regarding the tamoxifen treatment and the proposed role of the retromer in shiga toxin trafficking, the addition of controls and the analysis of the effect of tamoxifen on organelle pH. While I agree with the reviewers that analysing of >15 cells for the observed effects is at the low end, I accept your argument that the analysis fulfills Power analysis criteria. I also consulted with an expert from the field who thinks that your work merits publication at this stage. However, before moving towards acceptance of your article, a few changes should still get introduced:

1. I still think you cannot properly conclude that STx2 gets "re-routed" in absence of Rab2a or Vps39. The STx2 levels are clearly reduced as compared to WT cells, but I think that there may be alternative explanations for the reduced levels. Also, the control condition for the Leu/Pep treatment equally shows accumulation of STx2 outside of the Golgi (Fig 3M).

=> Please compare the STx2 levels side-by-side in Rab2a deficient and WT cells +/- Leu/Pep treatment and add the images for the Rab7DN assay (showing accumulation of STx2 outside of the Golgi) to further support your conclusions. Please also down-tone the proposed re-routing.

2. The control conditions in figure 4D and Fig 3 (STx2B/Rab7 stainings after 45 minutes) look quite different (though this could be due to DMSO treatment versus control RNAi treatment). Please mention this in the manuscript text.

3. Please address the following formatting requests:

- please list 10 authors et al in the reference list

3rd Editorial Decision

7 June 2019

Thank you for submitting your Research Article entitled "Tamoxifen blocks retrograde trafficking of Shiga toxin 1 and 2 and protects against lethal toxicosis". I appreciate the introduced changes and it is a pleasure to let you know that your manuscript is now accepted for publication in Life Science Alliance. Congratulations on this interesting work.